# Geospatial and Temporal Associations between Increases in Opioid Deaths, Socioeconomics, and Rates of Sexually Transmitted Infections in the Northeast United States 2012–2017

**DOI:** 10.3390/ijerph18010062

**Published:** 2020-12-23

**Authors:** Matthew R. Drewes, Jamison Jones, Emily Nelson Christiansen, Jordan P. Wilson, Brian Allen, Chantel D. Sloan

**Affiliations:** Department of Public Health, Brigham Young University, Provo, UT 84602, USA; mdraeves@gmail.com (M.R.D.); jjones37@luc.edu (J.J.); emi.christiansen3@gmail.com (E.N.C.); jordanpiwilson@gmail.com (J.P.W.); briangregoryallen@gmail.com (B.A.)

**Keywords:** HIV, gonorrhea, chlamydia, opioid, fentanyl, spatiotemporal

## Abstract

With the introduction of fentanyl to illegal markets in 2013 and an overall rise in rates of synthetic opioid use, opioid-related deaths have increased significantly. A similar trend has been observed for sexually transmitted infections, homicides, and poor mental health outcomes. In this paper, we explore the spatiotemporal relationship between opioid death rates and sexually transmitted infection (STI) rates in counties from the Northeast region of the United States between the years 2012–2017. We hypothesized that rates for gonorrhea, chlamydia, and human immunodeficiency virus (HIV) would all be positively associated with opioid death rates and that there would be a similar association between the STI rates and later time periods relative to earlier time periods. A negative binomial mixed-effects regression model was employed to assess these associations. Contrary to the study hypothesis, opioid death rates were not found to be significantly associated with the STI rates after accounting for other demographic and socioeconomic variables, with the exception of opioid deaths and gonorrhea in urban counties. Additionally, the regression demonstrated a significant association between infection rate and time period beyond the included socioeconomic variables and opioid deaths. Overall, this study indicates that declining sexual health outcomes may parallel rising opioid death, though both trends may be explained by similar underlying factors related to time period.

## 1. Introduction

Nearly 200,000 people died from opioid-related overdoses in the United States from 2010 to 2016 [1]. Within one year, 2014–2015, there was a 72.8% increase in fatal overdoses related to synthetic opioids nationwide [1]. The increase is primarily attributed to fentanyl and other synthetic drugs among heroin users, who one study found to be 3.9 times more likely to report nonmedical use of opioids during the prior year than those who do not use heroin [2]. This observation demonstrates how the majority of those using synthetic opioids began by misusing prescription opioids [3]. Fentanyl is up to 50 times as potent as heroin, and is often mixed with heroin in unknown quantities [4].

Areas along the eastern seaboard of the United States experienced some of the most rapid increases in deaths due to fentanyl [3]. Interstate 95, which runs from Maine to the southern tip of Florida, is a major route for illegal opioid distribution. Several northeastern states that are located along this corridor, including Connecticut, Maine, Maryland, Massachusetts, New Hampshire, New York, and Rhode Island were among the states exhibiting a statistically significant increase in synthetic opioid overdose from 2014 to 2015 [5].

Studies suggest a possible relationship between the rise in opioid related deaths and the coinciding increase in the incidence of sexually transmitted infections (STIs) [6,7,8]. For example, annual gonorrhea incidence in the United States reached over 580,000 in 2018 [9]. Several hypotheses as to the relationship between opioid use and STI incidence have been put forward. Needle sharing and increased high-risk sexual activity are found in communities with high levels of opioid abuse [10]. Rates of opioid deaths and STIs are both higher in areas with lower overall educational attainment, increased rates of violence, and lower socioeconomic status [11,12,13].

The goal of this study was to investigate whether the relationship between the rise in opioid death rates from 2012 to 2017 in the Northeastern United States was associated with changes in rates of STIs. Essentially, did the introduction of synthetic opioids prove to disrupt populations over geography and time in a way that may have likewise increased STI rates over geography and time? If so, was the relationship dependent on other variables (poverty, race, education)? We hypothesized that opioid deaths were positively correlated with an increased incidence of STIs, when controlling for other factors. 

## 2. Materials and Methods

### 2.1. Data

Demographic, homicide, and STI data were obtained from every county (where available), in Maryland, Washington D.C., Delaware, Pennsylvania, New Jersey, New York, Connecticut, Rhode Island, Massachusetts, Vermont, New Hampshire, and Maine between 2012 and 2017. Demographic data were obtained from the American Community Survey’s 5-year summary as of 2017 [14]. We obtained homicide data from the CDC Wonder detailed mortality database for 2012–2017 [15]. Homicides included incidents recorded under the “Homicide” subcategory in the “Injury Intent and Mechanism” categories and featured cause of death codes U01.4, X93-X95 (homicide by firearm) and UO1.0-U01.3, U01.5-U01.9, U02, X85-X92, X96-Y09, and Y87.1 (homicide by other and unspecified means). Opioid death data were also obtained from CDC Wonder database for “drug poisonings (overdose) unintentional, suicide, homicide and undetermined,” corresponding to codes X40-44, 60-64, 85, and Y10-14. With ICD-10 codes for “opium, heroin, other opioids, methadone, other synthetic narcotics, and other unspecified narcotics,” corresponding to codes T40.0, T40.1, T40.2, T40.3, T40.4, and T40.6. STI data (including gonorrhea, chlamydia, and human immunodeficiency virus (HIV)) obtained from the CDC NCHHSTP AtlasPlus [16]. Data were retrieved by selecting HIV, HIV diagnoses at the county-level from 2012–2017.

We aggregated the data within each of three time periods: 2012–2013, 2014–2015, and 2016–2017. Aggregating the data was required in order to avoid large numbers of suppressed counts due to low incidence in a single year. Year was coded as a continuous dummy variable for inclusion in the model. Socioeconomic variables, namely population density, poverty rate, percent of the population who were Hispanic, African American/Black (hereafter referred to as African American), and education levels (percent with a bachelor’s degree) were downloaded from the ACS 5-year summary from 2017. We chose to use the static rather than the time-variable data for the socioeconomic variables after verifying that they were relatively constant from 2010 to 2017. In contrast, opioid deaths, homicides, and STI rates were allowed to change over time. It should be noted that any time the word “rate” is mentioned hereafter, it refers to the number of cases per 100,000 persons in the population. 

There were 245 counties within the selected northeastern US states. Of these, data were divided into rural and urban subgroups, as the progression of both the opioid epidemic and rates of STIs were hypothesized to be different in rural vs. urban areas. Population density was calculated by dividing county population by area and is measured in persons per square kilometer and mile. Urban counties were defined as those with a population density equal to or greater than 518 persons per km^2^ (200 persons per mi^2^). Rural areas were defined as counties with a population density of less than 518 persons per km^2^ (200 persons per mi^2^). This cutoff allowed incorporation of more counties with a mix of rural land and major cities into the urban dataset that would otherwise be narrowly missed by the conventional measurement of 500 persons per square mile. Such cities included Burlington VT (≈300 persons/mi^2), Harrisburg PA (≈490 persons/mi^2) and Scranton PA (≈460 persons/mi^2). 

### 2.2. Statistical Analysis

As is evident from the histograms provided in Appendix A, all STI rates exhibited some degree of right skew, with gonorrhea and HIV featuring the strongest skew. In such cases, the mean of the sample exceeds the median. For example, the standardized mean–median spread for the urban counties was only about 0.27 for chlamydia, relative to 0.31 for HIV and 0.37 for gonorrhea. For the rural counties, the standardized spread for chlamydia was 0.15, relative to 0.27 for HIV and 0.30 for gonorrhea. 

We employed a negative binomial mixed-effects regression model to describe the association of gonorrhea, chlamydia, and HIV with opioid death rates and sociodemographics. We selected this model over alternatives for several reasons. First, our response variables (rates of STIs) exhibited mild to strong skewness (Figure A1 and Figure A4). Negative binomial regression is a generalization of Poisson regression that relaxes the assumption of equal mean and variance [17]. In order to account for the repeated measurements from each county, we additionally added a mixed effects term, using the glmbr.nb function in the lmer package in R (v. 4.0.3).

The frequency of cases was offset with the log of the total population in the county in 2015, as downloaded from data.census.gov. The model is structured as follows for each sexually transmitted infection: 

### 2.3. Rural and Urban Models

Due to the lower number of rural counties with complete data, we selected a simpler and more parsimonious model for rural analysis. Given the research question, the rural model was considered our “base model” upon which further variables were added in the urban model. Equation (1) provides the rural model specification. The urban model was expanded to also include race, ethnicity, education, and homicide as explanatory variables.
Cases ~ Year + Opioid Death Rates + PopDens + Poverty + (1|County) + offset(log(TotalPop))(1)

Equation (2) lists all possible explanatory variables that were considered during model fitting, but only those beyond the base model that improved the model, as determined by a lower Akaike information criterion (AIC), were kept. Our negative binomial mixed-effect regression model was fit using maximum likelihood estimation. For each explanatory variable for each STI and rural/urban type we report the exponentiated estimates, 95% confidence intervals, and statistical significance. We used a Moran’s I test to determine if spatial autocorrelation was present in the model residuals.
Cases ~ Year + Homicides + Opioid Death Rates + Pop_Dens + Poverty + Black + Hispanic + Bachelors + (1|County)+ offset(log(TotalPop))(2)

## 3. Results

We divided the counties into subgroups for analysis as shown in Figure 1, excluding counties for which key variables were missing. Of the rural counties, 89 were missing HIV data, of which 13 lacked opioid data. Thus only 27 counties (21%) had data available for an analysis of rural HIV. As only the most populous rural counties had HIV data, we determined that running a model with only these counties would not be truly representative of our rural variable, and thus a model was not created for the HIV subgroup. In contrast, only three rural counties were missing gonorrhea and chlamydia data. Opioid death data were not available for 73 rural counties, resulting in 53 available for inclusion in the gonorrhea and chlamydia analysis (41%). The presence of available data on homicide, while not required for the rural counties, was required for use in urban county analysis. Of the urban counties, nine were missing HIV data and 36 of those lacked opioid death and homicide data, resulting in 71 urban counties (61%) selected for HIV analysis. There was one urban county missing gonorrhea and chlamydia data, and 52 urban counties were further missing opioid and homicide data. Thus 63 urban counties (54%) were available for the gonorrhea and chlamydia analyses.

A descriptive analysis of explanatory variables for all counties, divided by study period (2012/2013, 2014/2015, and 2016/2017) is shown in Table 1. As data for percent African American, Hispanic, bachelor’s degree, and poverty came from a 5-year average from 2012–2017, those numbers are identical across all time windows. Opioid rates are available for all time windows, and homicide rates are also available for all time windows but only in urban counties. 

Table 1 shows that both urban and rural counties saw rises in average opioid overdose deaths. Between 2012/2013 and 2016/2017, average opioid death rates in rural counties increase by 11.9 deaths per 100,000, while rates in urban counties rise by 13.0 deaths per 100,000. In urban counties, average homicide rate does not seem to show a trend across all windows, decreasing 0.23 homicides per 100,000 from 2012/2013 to 2014/2015 and increasing 0.34 homicides per 100,000 from 2014/2015 to 2016/2017. 

Comparing rural and urban counties, African American population for urban counties was 8.16% higher than that of rural counties (3.14% rural vs. 11.3% urban) across all time windows. Urban counties also had an average Hispanic population 7.91% larger than that of rural counties (2.79% rural vs. 10.7% urban) across all time windows. Poverty was also more prevalent among urban counties, with the average percent living below the poverty threshold being 8% more in urban areas than in rural (11.3% urban vs. 3.3% rural) across all time windows. The trend prevailed for percent of the population with a bachelor’s degree, with 12.7% more holding bachelor’s degrees in urban areas than rural (35.3% urban vs. 23.6% rural). Summary statistics for the dependent variables within each of the time periods are also given for reference in Table 1. The trends mentioned above can also be visualized in Figure 3.

The geospatial distributions of the explanatory and dependent variables across all years are shown in Figure 2. Figure 2A demonstrates that chlamydia is generally distributed throughout all counties, with slightly higher rates in urban counties. Figure 2B,C shows high incidence of gonorrhea, and HIV primarily in urban areas along the I-95 corridor (between Washington DC and Boston) and other urban areas such as upstate New York and Pennsylvania. Additional representations of data can be found in Figure A1.

Tests of spatial autocorrelation among explanatory and response variables revealed that opioids, gonorrhea, chlamydia, percent African American, percent Hispanic, percent of population with a bachelor’s degree or higher, and percent of the population below poverty all showed significant geographic clustering. The only variable that did not show clustering was HIV incidence rates. Cross-correlations were compared between explanatory variables to determine whether variables should be omitted to avoid modeling problems associated with multicollinearity. All such correlations were found to be less than 0.8 and greater than −0.8, and no variables were omitted for this reason (Figure A3).

Distribution patterns in the sociodemographic, opioid, and STI data were similar between urban and rural counties. Percentages for African American and Hispanic population segments exhibited strong right skew, whereas an approximate bell curve was seen in education and poverty levels. Right skew was also present in opioid deaths and homicides. These data are consistent with current sociodemographic patterns in the Northeast United States (see Figure 2 and Figure 3). For example, most rural counties had small minority populations, while urban counties have larger minority populations. Education and poverty levels tended to be approximately normal in distribution. Opioid death and homicide rates are higher in more populated counties, while counties with smaller populations tend to have lower opioid death and homicide rates. For additional visualizations of sociodemographic data, refer to Figure A2, Figure A5 and Figure A6.

The fixed effects estimates from the rural model are shown in Table 2. No significant associations were found between STIs and opioid death rate per 100,000 in the rural population. However, statistically significant associations were found between gonorrhea incidence and time period (=1.44), gonorrhea incidence and population density (=1.46), gonorrhea and percent below poverty (=1.29), and chlamydia and percent below poverty (=1.18). For additional visualization of opioid death rate compared to other variables see Figure A7.

These estimates are in terms of odds ratios and are interpreted as multiplicative effects. For example, the estimate for the percentage below poverty variable in the model for gonorrhea (=1.29) is interpreted as an estimated 29% increase in the odds of gonorrhea rate in the population for every additional percent of the population that is living below poverty. There was no statistically significant autocorrelation detected in the model residuals.

The results of the model for urban counties are shown in Table 3. A significant relationship was found between opioid death rate and gonorrhea (= 1.11). There was no significant relationship found between opioid death rate and chlamydia or HIV. Statistically significant relationships were found between gonorrhea incidence and time period (= 1.11), homicide rate (= 1.28), population density (= 1.11), percent below poverty (= 1.20), and percent African American (= 1.21). There were statistically significant relationships between chlamydia incidence and time period (= 1.08), percent below poverty (= 1.20), and percent African American (= 1.20). There were also significant relationships between HIV incidence and time period (= 0.93), population density (= 1.27), percent African American (= 1.51), and percent Hispanic (= 1.19). Percent Hispanic was not included in the models for gonorrhea or chlamydia due to it not improving the model as defined by a lower AIC. Percent with a bachelor’s degree was dropped from all the models because it likewise did not improve model fit.

## 4. Discussion

Our study found that the relationship between gonorrhea and opioid deaths was significant only for urban counties after controlling for the previously specified covariates. This association is positive, as we originally hypothesized. For example, Philadelphia County, PA, ranks among the top three counties for STI rates but among the lowest for opioid death rate. This is inconsistent with reporting in other counties such as Baltimore City County, MD, which rank high for both opioid and STI rates (see Figure A8).

STI rates were not significantly associated with opioid-related death in the urban counties for chlamydia or HIV infection after controlling for demographic and socioeconomic variables, nor was there a significant relationship between chlamydia and opioid-related death in the rural counties. Within this time period, there was a significant increase in gonorrhea and chlamydia infection rates in both urban and rural communities. A similar significant increase was absent for HIV infection in this time period [18].

Our study also highlighted health disparities in the Northeastern United States. The STIs had significant associations with population density, percent African American, and percent below poverty after controlling for other variables in the model. Additionally, our study showed that many rural areas have been significantly impacted by the rise in opioid deaths, and studies have shown that access to care and treatment such as opioid-substitution therapy is limited in these areas. For example, it was found that less than 60% of the counties most vulnerable to HIV outbreaks (127 of 220) had a buprenorphine prescriber in 2018, according to Substance Abuse and Mental Health Services Administration registries [19].

The lack of a consistent reporting standard for opioid deaths across counties, along with underreporting of STIs in rural counties, is a major limitation of our study. The breakdown presented in Figure 1 shows that most of the counties excluded from our analysis were omitted due to missing opioid fatality data. Such omission introduces a possibility for selection bias which may affect the analysis results. For rural counties, we found that county observations excluded from the analysis differed from those included in the analysis in their average STI rates (Gonorrhea, Chlamydia, HIV: 25.3, 243.1, and 106.2 for excluded county observations vs. 35.3, 272.7, and 57.7 for included county observations) and in their average population density (72.8 for excluded observations vs. 118.6 for included observations). All other covariates were comparable. County observations excluded from the urban gonorrhea and chlamydia analyses also differed from included county observations somewhat in terms of average STI rates (Gonorrhea, Chlamydia, HIV: 54.3, 284.6, and 71.8 for excluded county observations vs. 86.0, 381.0, and 119.7 for included county observations), population density (1513.5 for excluded observations vs. 2964.2 for included observations), average homicide (7.4 for excluded observations vs. 4.7 for included observations), as well as average percent African American and percent Hispanic (9.5 and 6.7 for excluded observations vs. 11.4 and 11.1 for included observations, respectively). Poverty levels were comparable between the two groups. Differences between excluded vs. included county observations for the urban HIV analysis showed a similar trend with lower average STI rates, higher average homicide, and lower African American and Hispanic population percentages.

These differences point to the presence of fundamental differences between counties that are missing opioid data and counties that have such data available. Counties without opioid data tend to have smaller population density and lower reported rates of STIs (with the exception of HIV in rural areas where the rate was significantly higher in counties with missing opioid data), a higher proportion of nonminority residents, and higher rates of homicide in urban areas. Though the effects of these underlying differences on our estimates may be nuanced by the context of including multiple covariates in the analysis, it is our immediate thought that the estimates corresponding to the time period and poverty variables in the rural analyses may be overestimates and that the estimates for time period, poverty, and homicide may be overestimates in the urban analyses.

Lack of reporting standard for data related to opioid abuse may be attributed to several factors. Rural areas have less comprehensive sex education, are less likely to utilize public syringe exchange programs, and are more likely to stigmatize STI testing [20]. Reporting practices for opioid deaths vary by county. One example of how data collection practices may bias results is in Philadelphia, which is high on many of the metrics we investigated but low on the number of opioid deaths. Philadelphia’s low opioid death rate seems unlikely considering how other counties such as Baltimore, Maryland, rank similarly to Philadelphia for most of the included variables.

The relationship between time period and STI incidence suggests that there are other temporal variables unaccounted for in our model. These may include other socioeconomic factors such as homelessness, that act as confounders or more clearly show the relationship between STIs and opioid deaths. Lastly, there are potential differences between the counties included in our models and those (mostly rural) counties excluded due to lack of available data. It is important to note that our results may not extend to counties whose publicly reported health and crime data are not available for one reason or another (e.g., censoring due to privacy and anonymity concerns).

Improvement in reporting and recording standards for opioid deaths and STI rates is a known area of concern for many state and county public health systems. In the interim, our study demonstrates that there are likely spatial and temporal variables influencing the relationship between the rise of opioid deaths and the rise of STIs in counties across the northeastern United States.

## 5. Conclusions

Whether declining sexual health outcomes are directly related to rising opioid death is unclear. The results of this study support the notion that sexually transmitted infection rates have been increasing with time over the last decade. It is plausible that trends in both STI rates and opioid use may be explained by similar underlying factors without being directly related to one another. More research is needed to further explore this connection.

## Figures and Tables

**Figure 1 ijerph-18-00062-f001:**
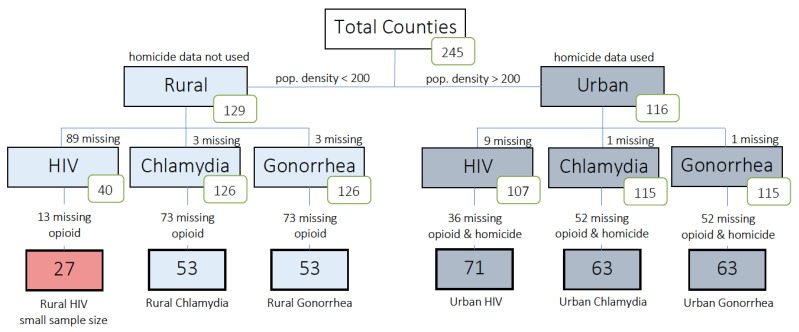
Flowchart showing the inclusion and exclusion criteria to create models of sexually transmitted infections for rural and urban northeast US counties. Counties were first separated into rural and urban groups by population density (200 persons per sq. mi.), then further separated by sexually transmitted disease—HIV, Chlamydia, or Gonorrhea. An unrepresentative and sparse dataset led to the exclusion of rural HIV data from the model.

**Figure 2 ijerph-18-00062-f002:**
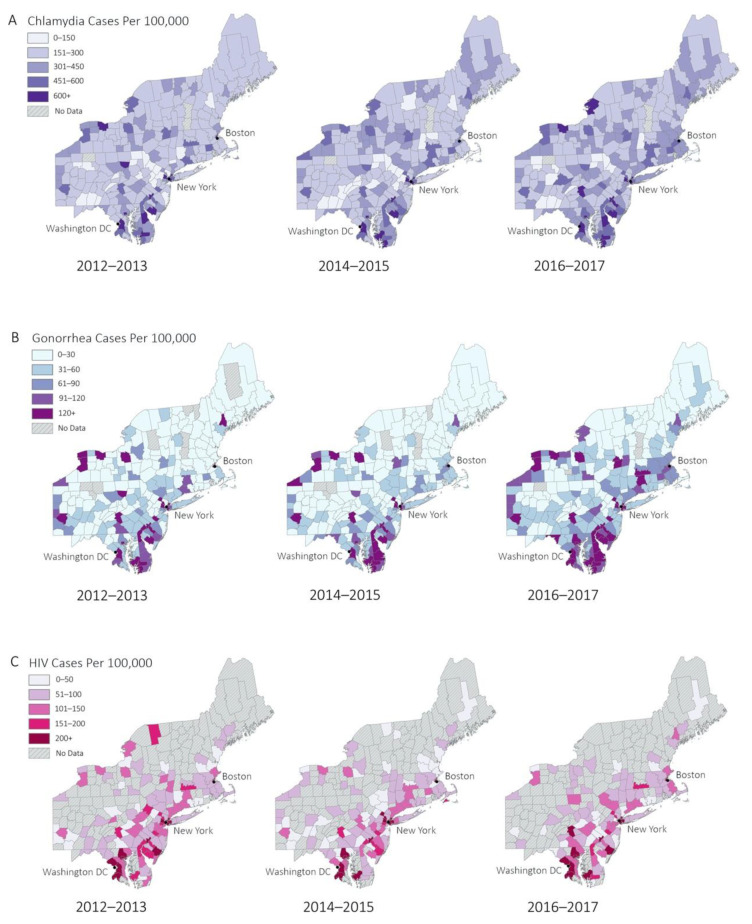
(**A**–**C**). Geographic distribution of dependent variables over all time windows. (**A**) Chlamydia cases per 100,000, (**B**) gonorrhea cases per 100,000, (**C**) HIV cases per 100,000. Colored regions on map represent county-level data. Counties with no data (either unrecorded or suppressed due to low case counts) were given a neutral (gray) color. Low rates of sexually transmitted infection are represented by lighter tones, while high rates are represented by darker tones.

**Figure 3 ijerph-18-00062-f003:**
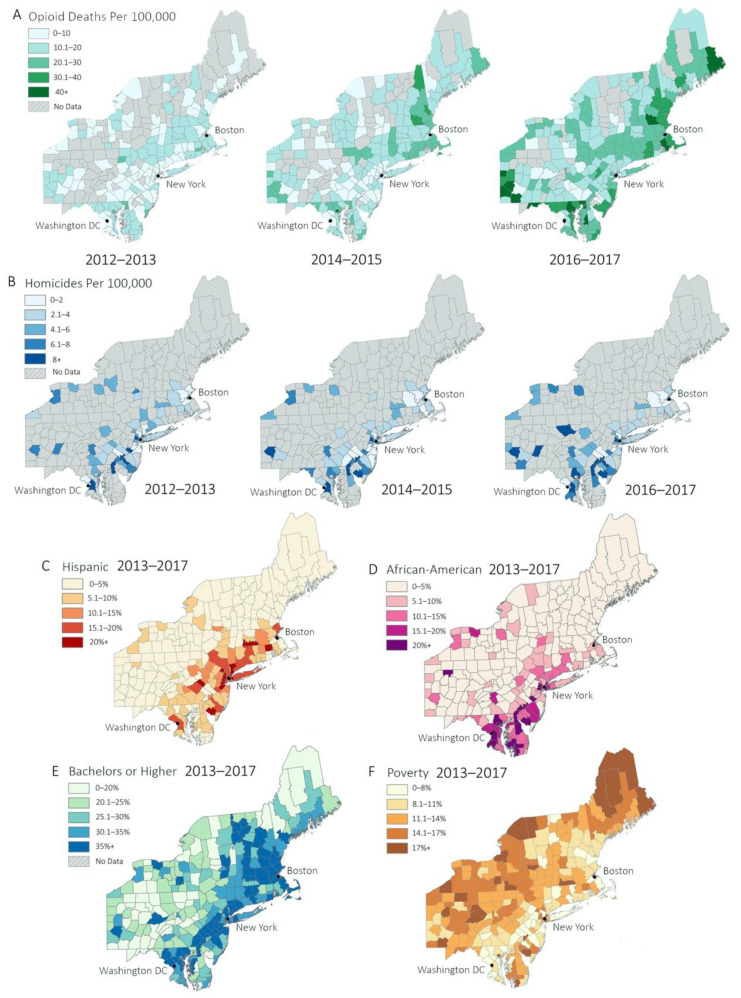
(**A**,**B**): Geographic distributions of independent variables over all time windows. (**A**) Opioid deaths per 100,000, (**B**) homicides per 100,000. (**C**–**F**): American Community Survey 5-year summary results from 2013–2017. (**C**) Percent of population Hispanic, (**D**) percent of population African American, (**E**) percent of individuals who have received a bachelor’s degree or higher, (**F**) percent of population living below the poverty line. Counties with no data were given a neutral (gray) color. Low rates of all variables are represented by lighter tones, while high rates are represented by darker tones.

**Table 1 ijerph-18-00062-t001:** Descriptive statistics of explanatory and model-dependent variables by year. Data is preScheme 100. persons, while the symbol “%” means percentage of the total population.

		Rural			Urban	
Variable	2012/2013	2014/2015	2016/2017	2012/2013	2014/2015	2016/2017
Death rate due to opioid overdose	11.8 (3.2)	18.7 (6.5)	23.7 (9.5)	10.7 (4.8)	14.3 (7.3)	23.7 (12.0)
Homicide rate	-	-	-	4.83 (3.4)	4.60 (3.1)	4.94 (3.3)
% African American	3.14 (5.4)	“	“	11.3 (11.7)	“	“
% Hispanic	2.8 (2.4)	“	“	10.7 (9.3)	“	“
% Below poverty	3.3 (3.1)	“	“	11.3 (4.6)	“	“
% Bachelor’s degree or above earned	23.6 (7.6)	“	“	35.3 (9.8)	“	“
Gonorrhea rate	23.1 (25.1)	25.3 (33.9)	34.8 (33.2)	73.4 (70.6)	73.6 (68.8)	102 (92.6)
Chlamydia rate	244 (93.9)	247 (104.0)	261 (106.0)	354 (210.0)	358 (193.0)	405 (205.0)
HIV rate	87.4 (48.4)	68.9 (58.8)	80.3 (74.1)	127 (126.0)	106 (93.5)	117 (89.2)

**Table 2 ijerph-18-00062-t002:** Estimates of fixed effects from the negative binomial mixed model regression analysis for each of the dependent variables for rural counties (gonorrhea, chlamydia, and HIV rates). Rural areas were defined as counties with a population density of less than 518 persons per km^2^ (200 persons per mi^2^). We did not include HIV in this analysis due to the biased nature of the data available for only the most populous counties fitting into the “rural” categorization.

	Gonorrhea (N = 53)		Chlamydia (N = 53)	
		95% CI		95% CI
Time period	1.44 ***	1.26–1.66	1.05	0.97–1.15
Opioid death rate	0.95	0.84–1.07	0.99	0.92–1.06
Population density	1.46 ***	1.25–1.71	1.09	0.99–1.19
Percent below poverty	1.29 ***	1.11–1.51	1.18 ***	1.08–1.28

* Indicates statistically significant results, *** <0.001.

**Table 3 ijerph-18-00062-t003:** Estimates of fixed effects from the negative binomial mixed model regression analysis for each of the dependent variables for Urban Counties (gonorrhea, chlamydia, and HIV rates). Urban counties were defined as those with a population density equal to or greater than 518 persons per km^2^ (200 persons per mi^2^).

	Gonorrhea (N = 63 Counties)		Chlamydia (N = 63 Counties)		HIV (N = 71 Counties)	
		95% CI		95% CI		95% CI
Time period	1.11 **	1.04–1.19	1.08 ***	1.04–1.12	0.93 ***	0.89–0.97
Homicide rate	1.28 **	1.07–1.43	1.07	0.99–1.15	0.98	0.91–Inf
Opioid death rate	1.11 *	1.02–1.20	1.00	0.96–1.05	1.03	0.98–1.08
Population density	1.11 **	1.02–1.21	1.04	0.99–1.08	1.27 *	1.03–1.56
Percent below poverty	1.20 **	1.07–1.36	1.20 ***	1.12–1.28	1.07	0.97–1.18
Percent African American	1.21 **	1.06–1.39	1.20 ***	1.11–1.30	1.51 ***	1.39–1.66
Percent Hispanic ^a^	NA	NA	NA	NA	1.19 ***	1.10–1.29

* Indicates statistically significant results, * <0.05, ** <0.01, *** <0.001. ^a^ Percent Hispanic was not included in the models for gonorrhea or chlamydia due to it not improving the model as defined by a lower AIC. Percent with a bachelor’s degree was dropped from all models because of lack of improvement of the AIC.

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
