# Peer review of "Geospatial and Temporal Associations between Increases in Opioid Deaths, Socioeconomics, and Rates of Sexually Transmitted Infections in the Northeast United States 2012–2017"

_ijerph, 2020, doi:10.3390/ijerph18010062_

Round 1

Reviewer 1 Report

The authors investigate the important question of a correlation between increased rates of opioid use and increased rates of STIs. They collected dates across 246 counties in the northeastern US region. The main research questions were the association of the opioid and STI rates in the counties, as well as the time period (2012-2013, 2014-2015, 2016-2017); corrected for several covariables. While the research question is interesting and timely, there are several flaws in the methods, results and presentation of the manuscript which should, in my opinion, be addressed before publication:

1) The authors did a good job in properly selecting the model for the research question, namely negative binomial. They explain that due to the right skewness of the STI rates, Poisson regression is not a good choice. I fully agree with this choice of the model. However, in my opinion, because of the same argument (skewness of the STI rates), T-tests should not be used for comparison. It is not clear what was tested in Table 1 and 2 (last column says "t-test"), and also, how this t-test was applied for the 6 columns (or was it tested on aggregated data over all 3 time frames?). The authors should clarify this. It is also hard to interpret the results in tables 1 and 2. The Figure legends need to be expanded.

2) In Figure 1, the authors present a flow chart of the study selection. There seem to be several typos in the flow chart (e.g. chlamydia 129 for rural: should be 126?). The authors argue that having HIV data of 27/129 rural counties is not enough for further analysis. I am missing a proper power calculation and proper argumentation of why this information should be excluded.

3) It is not clear how the temporal relationship was investigated. It seems (although not explained properly) that the time variable was included as a categorical variable (2012/2013, 2014/2015, 2016/2017). I am wondering whether this increase over time could rather be introduced in the form of a slope, or similar, to capture the increase over time, rather than having 3 categories. (with categorical variables, the model does not "know" that 2012/2013 was before 2014/2015, was before 2016/2017; these are just 3 categories). Since the temporal association was one of the main research questions, this parts need to be improved.

4) In general, the reading flow is not easy and it is hard to get the main message. The results section of the manuscript could be re-arranged and shortened to increase readability. In my opinion, there are too many tables and figures in the main manuscript. E.g. Table 5 is very detailed and does not add much to the main results. In general, the analysis of single counties could be moved to a supplementary table to shorten the main manuscript.

Minor comments:

1) It is hard to understand Figure A1: what are the x-axes? You display the distribution of STI rates of the 246 counties, right?

2) It is confusing to change between rate and number/100000, this could be made more consistent.

3) In general, the authors should expand on the figure legends. Especially for the Appendix.

Reviewer 2 Report

Thank you for the opportunity to review this interesting submission. This article requires substantial changes before it is suitable for publication. The theoretical backing is unclear as are the implications of the findings. The statistical analysis needs to be rethought and is also lacking clarity. I recommend that the authors attempt to refocus their analysis to more closely mirror their stated hypotheses. See specific comments below:

Major Comments

  • The JAMA article you cite in line 45 states: “‘We think it's not necessarily or directly related, [However,] we do see these connections between opioid and other drug use with these sexual behaviors and STD rates.’” It seems that your analysis echoes this statement with no mechanism for further uncovering this relationship. To suggest that your results say anything about individuals (which you rightly avoid) would be the ecological fallacy, so the most interesting interpretation you can obtain from these data is to echo what others are saying: that there may be a relationship between opioid use and STI rates, but you cannot determine its existence or direction on the individual level with your methods. This does not mean it is not a publishable result, but it reduces the potential impact of the work. This work would benefit from additional motivation as to why this association might be relevant at the county level and what implications its potential existence might have.
  • Both of your stated hypotheses (lines 46-50) concern opioid use but your analysis uses opioid deaths. You need a strong justification for using opioid deaths instead of opioid use, ideally besides data availability. Theoretically, why would there be an association with deaths – is it just that more use -> more deaths so deaths are proxy for use? If you have opioid use data, I suggest that you use this instead to mirror your hypotheses.
  • Similarly, your motivation and hypotheses focus on changes in rates (lines 51-53) but you are modeling absolute amounts in your statistical analysis. You therefore can only make inferences on cross-sectional correlations and not how one change in rate impacts another.
  • I recommend that you do not round your rates to create counts (lines 109-111). I suggest using an offset for the number of individuals in each country and the number of deaths/cases observed as the outcome. This way, you are allowing the variance calculations to account for the relative populations of each county and are not introducing unnecessary noise through rounding.
  • You need to account for repeated observations for counties where data is measured during 3 different time periods by using a random-effects model (lme4::glmer in R) or GEE with a specified correlation structure. Otherwise, you are underestimating the standard error of your betas.

Minor Comments

  • Why was a population density of 200 persons per square mile chosen as a cut-off between rural and urban (line 241)? Needs a justification and/or citation.
  • Are the counties that you excluded from the regression analysis different from the counties that were included? What does this mean for interpretation? Would it be helpful to impute low values if they are excluded due to anonymity concerns?
  • I don’t think that you need to derive the mean and variance of the negative binomial distribution here (lines 112-120). It would be more helpful to see your actual fitted model described here with all its covariates.
  • Which variables were in your model when you compared the Poisson versus the negative binomial regression (lines 126-128)?
  • You say that the association between gonorrhea and opioids was significant only for rural counties in years 2016-2017, but from table 3, it seems that the beta for your opioid death rate term includes all years?

Reviewer 3 Report

The authors evaluated the rates of sexually transmitted diseases correlated with opioid death rates. The authors explained why they used a negative binomial regression model for this study. This paper explained the merits and limitations of the correlation study. Do authors see any correlation between opioid death rates and other STIs like Herpes, HPV?

Author Response

Response to Reviewer 3 Comments

Point 1: Do authors see any correlation between opioid death rates and other STIs like Herpes, HPV?

Response 1: That would have been a very interesting correlation to investigate. Unfortunately, we were not able to find sufficient data at the county level to measure other STIs. 

Round 2

Reviewer 1 Report

The authors clarified all points from the previous revision.

Author Response

Thank you for your feedback!

Reviewer 2 Report

I commend the authors for quickly making large improvements to this manuscript. I’d also like to commend the authors for working hard to publish largely null results, as they are also of scientific importance. There are some minor changes to consider before publication.

Comments

  • Please try to be a bit more direct in your writing. For example: "These features of the data are consistent with an understanding of the current sociodemographic climate in the United States, specifically the Northeast." -> "These data are consistent with current sociodemographic patterns in the Northeast United States."
  • In figure 1, I’d replace “high error” with “small sample size” because “high error” could be misinterpreted as the data has errors in it.
  • Table aesthetics: You could combine tables 1 and 2, I see no reason they should be separated. Also, might be easier to show the increases over time if the first three columns were Urban periods 1-3 and the last three columns were Rural periods 1-3 – easier for the eye to make comparisons. Could also introduce a new row above the three periods with just one “Urban” or “Rural” label depending on the periods, instead of repeating it for each column.
  • For figure 2, it might be helpful to specify whether there is no data or if the data were suppressed due to low case counts.
  • Your tests of spatial autocorrelation should be specified in the methods section.
  • “These estimates are in terms of odds (continuous variables) and odds ratios (categorical variables) and are interpreted as multiplicative effects.” – The continuous variables still return an odds ratio. It is the odds ratio of X+1 compared to X.
  • Table 3: shouldn’t “time period” have 2 rows since there are three groups being compared, with one group being the reference? Unless time period was made to be continuous, and if so, this should be specified as the earlier grouping suggested it would be categorical.
  • Table 3: Clarify what are the units for population density? The odds of gonorrhea increasing by a factor of 1.46 due to an increase of a single person per square mile seems high.
  • Discussion: any time you talk about a correlation, it would be best to specify that it is controlling for the specified covariates.
  • Discussion: “This association is positive, as we originally hypothesized, but may be due to differences in record keeping rather than representative of a true underlying relationship.” – I’d either be more specific here about “record keeping” or wait until the limitations paragraph.
  • Conclusion: “The results of this study support the notion that sexually transmitted infection rates have been increasing with time over the last decade, even when socioeconomic factors are taken into account.” – but the socioeconomic factors you control for are time-invariant, so controlling for these variables would not change an observed time trend.
  • Selection bias and missing data is the biggest challenge for your study and deserves proportional treatment. This could justify a supplemental table which has the characteristics of included counties compared to excluded counties. At minimum, the implications of this selection bias should be expanded in the discussion. What do you observe/expect to be different about your included and excluded counties? In which direction might this shift your estimates?
